# Expression of Tryptophan 2,3-Dioxygenase in Metastatic Uveal Melanoma

**DOI:** 10.3390/cancers12020405

**Published:** 2020-02-10

**Authors:** Mizue Terai, Eric Londin, Ankit Rochani, Emma Link, Bao Lam, Gagan Kaushal, Alok Bhushan, Marlana Orloff, Takami Sato

**Affiliations:** 1Department of Medical Oncology, Sidney Kimmel Cancer Center, Thomas Jefferson University, Philadelphia, PA 19107, USA; elink507@gmail.com (E.L.); Bao.Lam2@jefferson.edu (B.L.); Marlana.Orloff@jefferson.edu (M.O.); Takami.Sato@jefferson.edu (T.S.); 2Computational Medicine Center, Sidney Kimmel Cancer Center, Thomas Jefferson University, Philadelphia, PA 19107, USA; Eric.Londin@jefferson.edu; 3Department of Pharmaceutical Sciences, Jefferson College of Pharmacy, Thomas Jefferson University, Philadelphia, PA 19107, USA; Ankit.Rochani@jefferson.edu (A.R.); Gagan.Kaushal@jefferson.edu (G.K.); Alok.Bhushan@jefferson.edu (A.B.); 4College of Medicine, Drexel University, Philadelphia, PA 19129, USA

**Keywords:** uveal melanoma, metastatic uveal melanoma, liver metastatic, tryptophan 2,3-dioxygenase, TDO, tryptophan, kynurenine, LCMS, tumor microenvironment, immunotherapy

## Abstract

Uveal melanoma (UM) is the most common primary eye malignancy in adults and up to 50% of patients subsequently develop systemic metastasis. Metastatic uveal melanoma (MUM) is highly resistant to immunotherapy. One of the mechanisms for resistance would be the immune-suppressive tumor microenvironment. Here, we have investigated the role of tryptophan 2,3-dioxygenase (TDO) in UM. Both TDO and indoleamine 2,3-dioxygenase (IDO) catalyze tryptophan and produce kynurenine, which could cause inhibition of T cell immune responses. We first studied the expression of TDO on tumor tissue specimens obtained from UM hepatic metastasis. High expression of TDO protein was confirmed in all hepatic metastasis. TDO was positive in both normal hepatocytes and the tumor cells with relatively higher expression in tumor cells. On the other hand, IDO protein remained undetectable in all of the MUM specimens. UM cell lines established from metastasis also expressed TDO protein and increasing kynurenine levels were detected in the supernatant of MUM cell culture. In TCGA database, higher TDO2 expression in primary UM significantly correlated to BAP1 mutation and monosomy 3. These results indicate that TDO might be one of the key mechanisms for resistance to immunotherapy in UM.

## 1. Introduction

Uveal melanoma (UM) originates from melanocytes in the iris, ciliary body, and choroid. The estimated incidence of UM is 5 per million in the United States [1], and between 2 to 8 cases per million in Europe [2,3]. Up to 50% of patients develop metastatic disease, with the liver being a predominant metastatic site in 70% to 90% of cases [4]. Despite advances in the treatment of primary UM, treatment options for metastatic UM (MUM) remain limited [4,5].

UM displays chromosome aberrations and gene mutations that correlate strongly with clinical outcomes. Loss of one copy of chromosome 3 (monosomy 3) and BAP1 mutation in primary UM is associated with an increased risk of metastasis and a poor prognosis [6]. A previous study on TCGA data found that the four subtypes of primary UM have distinct molecular signatures. Two subtypes are associated with monosomy 3 (M3) and chromosome 8q gain with poor prognosis, and the other two are associated with disomy 3 (D3) and chromosome 6p gain with better prognosis [7]. Despite the fundamental knowledge of the molecular phenotypes of UM, effective treatments for metastatic uveal melanoma (MUM) have not been developed.

The recent development of immune checkpoint inhibitors (ICIs) revolutionized immunotherapy for metastatic cutaneous melanoma (CM). Unfortunately, MUM is highly resistant to ICIs [8]. The reasons underlying the poor responses to immunotherapy in MUM remain speculative. One of the mechanisms by which tumors escape the immune system is through the upregulation and expression of immunosuppressive molecules. Our laboratory investigated programmed death-1 (PD-1) and programmed death-ligand 1 (PD-L1) expression levels in MUM tissues and reported low expression of PD-L1 in hepatic metastasis from UM, compared to metastasis from cutaneous melanoma [9]. PD-L1 can be expressed in metastatic UM cell lines in response to interferon-gamma (IFN-γ) and this phenomenon indicates a lack of activated T cells in MUM tumor microenvironment. Since tryptophan 2,3-dioxygenase (TDO) is predominantly expressed in the liver, we speculated that TDO encoded by the TDO2 gene might play a role in T-cell non-inflamed tumor microenvironment in the liver.

Under physiological conditions, the kynurenine (Kyn) pathway helps in carrying out metabolism of tryptophan (Trp) to manage various essential metabolic pathways of the human body, including NAD^+^ biosynthetic pathway and TCA cycle [10]. Further, Kyn pathway is now firmly established as one of the key regulators of immunity [11,12]. Two enzymes, TDO and indoleamine 2,3-dioxygenase (IDO), regulate the first and rate-limiting step of the Kyn pathway. IDO is expressed in many tumors and induced by IFN-γ to contribute to tumoral resistance to immune rejection [13]. In contrast, the liver constitutively expresses TDO and is one of the major organs for TDO-mediated metabolism of Trp to Kyn. Moreover, TDO has a higher capacity for Trp than IDO in the human liver [10]. In terms of the role of TDO in cancer, TDO is strongly expressed in various cancers including glioma and TDO-derived Kyn promoted glioma cell survival and migration thorough the aryl hydrocarbon receptor (AHR) activation in an autocrine/paracrine fashion [14,15,16].

Trp is an essential amino acid and is required for T cell and NK cell functions, whereas the metabolically-generated Kyn suppresses the proliferation of T cells and NK cells and makes them sensitive to apoptosis stimuli [12,15,17]. The catabolites of Trp in tumor microenvironment is responsible for immune response suppression and consequently prevents tumor cells from immunological rejection [18].

Hence, in the present work, we investigated the expression of TDO2 mRNA and TDO protein in hepatic metastasis tissue specimens, as well as cell lines collected from stage IV UM patients. Using liquid chromatography high resolution mass spectrometry (LC-HRMS), we aimed to confirm that TDO in MUM cells is functional and increases Kyn levels in culture media. We also investigated TDO2 mRNA expression in the TCGA database of primary UM and correlated it with a prognostic outcome. Our data suggest that TDO expressed by UM cells might be one of the key mechanisms for resistance to immunotherapy in UM.

## 2. Results

### 2.1. The Detection of (TDO) Protein by IHC in MUM Tissue Sections

We first investigated whether TDO protein is detectable in MUM tissues. Immunohistochemical (IHC) staining of TDO protein in 16 archived hepatic metastasis samples from stage IV UM patients show presence of TDO protein in all specimens (Figure 1A,D and Appendix A). The expression of TDO is detected in both melanoma and noncancerous areas of liver tissue in all patients (Figure 1 and Appendix A). The specificity of TDO antibody was confirmed by blocking peptides. In contrast, IDO is found to be negative in all MUM tissue sections (0/14) (Figure 1C,F and Appendix A). Interestingly, the intensity of TDO staining tended to be higher in metastatic melanoma cells themselves (Figure 1B,E and Appendix A), compared to that of surrounding normal hepatocytes (Figure 1 and Appendix A). The data obtained from IHC staining of MUM tissues indicate a special role of TDO in MUM.

### 2.2. RNA with in Situ Hybridizations in the MUM Tissue Sections

To further confirm the specificity of anti-TDO antibody, detection of TDO2 mRNA was carried out with the RNAscope method in the hepatic metastasis tissue sections. We used an in-situ hybridization assay to detect TDO2 mRNA within intact cells. As shown in Figure 2, melanoma cells and surrounding liver tissue show positive TDO2 mRNA expression (Figure 2a–c,f–h). Staining of TDO2 mRNA is much stronger in MUM cells (Figure 2c,h) compared to the surrounding liver tissues (Figure 2b,g). TDO protein is also positive in the area where TDO2 mRNA expression is positive (Figure 2d,i). The tissues were also stained with HMB45 to confirm melanoma cells (Figure 2e,j). These data confirm that TDO is dominantly expressed in MUM cells in the liver. In addition to the physiological presence of TDO in hepatocytes, enriched expression of TDO was confirmed in uveal melanoma cells.

### 2.3. The Expression of TDO and IDO in MUM Cell Lines

To test the enzymatic functions of TDO in MUM cells, the presence of TDO2 mRNA and TDO protein were first investigated in established MUM cell lines. As showed in Figure 3A, all tested MUM cell lines are positive for TDO2 mRNA. In contrast, two out of four cell lines are IDO1 mRNA clearly positive. The expression of TDO protein in MUM cell lines is also confirmed using Western blotting (Figure 3B and Appendix A). However, IDO protein expression is found to be negative in all metastatic UM cell lines. Once melanoma cells were stimulated with IFN-γ for 48 h, IDO protein became detectable (Figure 3B and Appendix A). The result indicates that TDO protein is constitutively expressed in the MUM cells, whereas MUM cells express IDO protein only when they encounter IFN-γ.

### 2.4. Expressions of TDO2 and IDO1 in Response to TNF-α and IFN-γ

It has been reported that various molecules are upregulated in tumor microenvironment in response to immunological attack against cancer cells. Especially, Th1 type cytokines, such as IFN-γ and TNF-α, are known to trigger upregulation of counter-attack molecules, including IDO and PD-L1. We tested the effect of inflammatory cytokines commonly found in tumor microenvironment including IL-1β, IL-6, IL-10, TNF-α, and IFN-γ on TDO2 expression on MUM cells. As shown in Figure 4A, TNF-α at a dose of 1 to 10 ng/mL increases the expression of TDO2 mRNA substantially in UM004 cell line, while TNF-α had little effect on the expression of IDO. IL-1β, IL-6, and IL-10 did not induce the expression of either TDO or IDO. When we examined the other three MUM cell lines, we found an increase in the TDO2 mRNA by 2–8 folds (compared to untreated MUM cell lines) at the dose of 10 ng/mL of TNF-α. Moreover, IFN-γ stimulates expression of IDO1 but does not induce TDO2 mRNA in MUM cells (Figure 4B). These data indicate that TNF-α, not IFN-γ, might stimulate upregulation of TDO2 mRNA by MUM cells in a tumor microenvironment.

### 2.5. Measurement of Kynurenine (Kyn) Metabolite Produced by MUM Cell Line

We further investigated whether TDO in MUM cells is functional. This is confirmed by measuring Kyn concentration in the culture supernatant from the UM004 cells. Since TNF-α upregulated TDO2 RNA expression, we investigated production of Kyn by MUM cells in response to TNF-α. In contrast, IFN-γ did not change the TDO2 RNA expression level; therefore, the influence of IFN-γ on Kyn production was not tested in this assay. Kyn levels in the supernatant from UM004 cells are significantly increased after 24 h culture (Figure 5). In addition, the addition of TNF-α further increases Kyn levels in the culture supernatant of UM004 cells (Figure 5), which is consistent with the upregulation of TDO2 expression by TNF-α (Figure 4A).

### 2.6. The Role of TDO2 RNA Expression in Primary Uveal Melanoma

TDO2 is located at 4q32.1 and ubiquitously expressed throughout all 33 cancer types included within the TCGA cohort (Figure 6A). While TDO2 RNA shows a range of expression profiles across the cancers, the UM cohort consisting of 80 primary UM displays the lowest median expression. We then analyzed the survival comparing the subjects with TDO2 RNA expression vs. those without TDO2 RNA expression. The expression of TDO2 RNA has no association with survival (*p* = 0.77) in this comparison. Within the UM dataset itself, less than 50% of primary uveal melanomas (34/80) have a detectable expression of TDO2 RNA (Figure 6B). While expression of TDO2 RNA is generally low in primary UM, we further examined if the expression is correlated to clinical or genomic attributes (Appendix A). For 34 primary UM patients with positive TDO2 RNA expression, we stratified them based upon clinical or molecular attributes and determined if these correlated to the TDO2 RNA expression. We found that patients with BAP1 mutations have increased expression of TDO2 (*p* = 0.007) compared to BAP1 wild-type. Similarly, patients with monosomy 3 (M3) primary uveal melanoma have increased TDO2 RNA expression (*p* = 0.001). We then stratified the expression of TDO2 RNA into four distinct cluster groups according to chromosomes 3, 6, and 8 copy-number aberrations [7]. Data were assessed to Cluster 1: Disomy 3 (D3) with enriched chromosome 6p; Cluster 2: D3 with chromosome 6p gain with partial chromosome 8q gain; Cluster 3: M3 with chromosome 8q gain; and Cluster 4: M3 with increased copy numbers of chromosome 8q gain. TDO2 RNA expression is higher in Cluster 4 which has been associated with UM metastasis and poor prognosis in TCGA database (Figure 6C).

Interestingly, when we focused on only patients whose tumor expresses TDO2 (*n* = 34), the survival of patients with high TDO2 RNA expression (those above the median) showed shorter survival compared to patients with low TDO2 RNA expression (those below the median) (*p* = 0.007) (Figure 7).

Taken together, these data suggest that in primary UM, TDO2 RNA expression is generally low (compared to other cancers), but the increased TDO2 RNA expression is associated with the poor prognosis markers, BAP1 mutations, and M3. More importantly, the degree of expression in TDO2-positive primary UM correlated to the survival of patients.

## 3. Discussion

Despite successful treatment with immune checkpoint inhibitors (ICIs) in metastatic cutaneous melanoma [19], the efficacy of these medications is limited in MUM [20]. Previously reported clinical trials with more than 30 MUM patients showed the response rates of anti-CTLA-4 antibody (ipilimumab) and anti-PD-1 antibodies (nivolumab and pembrolizumab) to be 0–7.7% and 3.6–5.8%, respectively [21]. A variety of mechanisms may account for such resistance to ICIs, including low tumor mutational burden and low tumor neoantigen expression [22]. Another unique characteristic of UM is that the majority of metastases first develop in the liver. The liver is highly specialized in the development of immune tolerance to food-derived antigens and provides MUM cells with a niche for survival by decreasing immunological attacks by the host immune system.

It has been reported that tumor-infiltrating T cells obtained from MUM were difficult to expand ex vivo. PD-L1 is not expressed in the liver metastasis from UM. The lack of PD-L1 expression in tumor tissues and low expansion of CD8 T cells suggest that the liver is a challenging immune microenvironment for tumor-specific T cells [23]. We hypothesized that one of the mechanisms that contribute to deficient T cell immune responses in the liver metastasis is TDO-mediated Trp catabolism, since the liver is rich in TDO. Conversion of Trp into Kyn (in presence of TDO) causes Trp starvation in the microenvironment which suppresses immune cell functions [24]. Apart from having immune suppression induced by Trp catabolism, Kyn binds to the aryl hydrocarbon receptor (AHR), a cytoplasmic transcription factor. TDO-derived Kyn suppresses antitumor immune responses and promotes tumor-cell survival and motility through the AHR in an autocrine/paracrine fashion [14]. Kyn induces formation of regulatory T cells (T regs) and the administration of a TDO inhibitor improves the function of dendritic cells (DCs) and decreases lung cancer metastasis in mice [14].

Much to our surprise, we confirmed that not only hepatocytes, but also MUM cells themselves, express TDO. We demonstrated the expression of TDO2 mRNA and TDO protein in tumor tissue specimens from MUM patients as well as MUM cell lines. This is the first demonstration of TDO2 mRNA as well as functional TDO protein in MUM tumors. Universal and constitutive expression of TDO in MUM cells suggest an important role of TDO in the survival and progression of MUM cells. Although the MUM cell line was able to express IDO in the presence of IFN-γ, IDO was shown to be negative in archived hepatic metastasis samples from stage IV UM patients. This indicates the important role of TDO in MUM in the liver to suppress immunological activities and potentially promote the progression of the disease.

It is interesting that there is no survival difference in primary UM patients with and without TDO2 RNA expression in TCGA data. On the other hand, a clear survival difference is indicated between patients with TDO2 high and TDO2 low primary UM. It could be speculated that TDO does not have any meaningful role in tumors that do not express TDO. There are many different immune suppressive mechanisms in cancer cells and a different immuno-modulatory mechanism would exist in primary UM without TDO expression. On the other hand, in UM with expression of TDO2, the degree of TDO2 mRNA expression would influence the outcome of patients. It would also be possible that primary UM without TDO2 expression starts expressing TDO in response to a factor in the tumor microenvironment. We identified that one of such factors could be TNF-α. This might be supported by the fact that TDO2 RNA is dominantly expressed in Cluster 4 primary UM, in which significant infiltration of immune cells are present. This possibility is further supported by our findings that all metastatic specimens that we tested were positive for TDO protein and TDO2 mRNA. TDO expression in normal liver tissue would provide additional protection to MUM cells from an immune attack by the host immune system.

The biological and clinical correlation with TDO has been tested in triple negative breast cancer (TNBC). Co-culturing of CD8^+^ T cells with conditioned media from TDO-positive TNBC cells decreased IFN-γ level and increased apoptosis of CD8^+^ T cells [25]. Higher expression of TDO in TNBC correlated with poor overall survival (OS) [16]. Publicly available data confirmed that higher expression of TDO2 RNA in breast cancers correlated with worse OS and decreased distant metastasis-free survival, which is inconsistent with our data on primary uveal melanoma with TDO2 RNA expression. However, there was no difference in OS and metastasis-free survival between these groups with IDO1 RNA expression [25].

The expression of TDO2 RNA from the TCGA database also shows a correlation between TDO2 mRNA expression and poor prognostic markers (BAP1 mutation and Monosomy 3 (M3)) in UM. It is also of note that M3 primary UM is associated with the presence of an inflammatory infiltrate and expression of IDO, CTLA4, and PD-L1 [7,26]. To date, the regulation of TDO2 (in comparison to IDO) expression has been poorly understood. Triple-negative breast cancer (TNBC) induced TDO2 mRNA in co-culture with TNF-α and IL-1β This increase is through NF-κβ activation [16]. Our data support this possibility and showed that when TNF-α, activator for NF-κβ, was added to the MUM cell culture, it further increased TDO2 mRNA in MUM cells. At this moment, we are not sure whether expression of TDO2 RNA in high risk primary UM is related to molecular or genetic alternation in UM cells or in response to an inflammatory tumor microenvironment, as seen in Cluster 4 UM. Further investigation is needed to answer this important question.

It is of note that the expression of IDO was associated with a poor prognosis in various malignancies. Therefore, the efficacy of the IDO inhibitor in combination with various drugs was investigated in ECHO301-310 trials. Unfortunately, the results of phase III trial (ECHO301) in melanoma indicated that IDO-inhibitor treatment, in combination with anti-PD-1 antibody, showed no improvement in progression-free survival compared to the anti-PD-1 antibody alone [27]. These results raise fundamental questions about the benefit of IDO inhibitors in cancer treatment. One of the reasons for trial failure could be explained by the fact that the tested IDO inhibitor does not cross-inhibit TDO. Endogenous expression of TDO in various cancers and in the liver may inhibit T cell activities in the tumor microenvironment and counteract the pharmacological effects of IDO specific inhibitors. In this regard, dual inhibitors of IDO and TDO might need to be tested in future clinical trials.

Despite potential benefits from TDO inhibitor, a large-scale clinical study has not been done, mainly due to toxicity concern. In a normal physiological condition, TDO is mainly expressed in the liver, which is very important for controlling excess Trp levels [28]. Since TDO performs important physiological functions, inhibiting TDO may have serious side effects. In this regard, inhibition of TDO for 3 months did not show a difference in the level of hepatic toxicity between untreated and treated mice [18]. Transgenic mice that did not express TDO did not exhibit any negative biological activities [29]. In addition to safety concerns, we speculate that dual inhibitors for TDO and IDO need to be considered due to compensatory roles of these enzymes. It is possible that tumor cells start upregulating IDO after TDO inhibitor treatment as a compensatory mechanism of Trp-Kyn pathway or in response to IFN-γ production by activated tumor-infiltrating T cells after blocking TDO.

Taken together, TDO2 expression in UM indicates poor prognosis, which is consistent with the results of other types of cancers [15]. The presence of TDO2 RNA and TDO protein in all tested MUM tissues indicate that TDO could be associated with the development and growth of metastasis in UM. Furthermore, the development of metastasis in the liver might have an additional advantage, since the liver is the major physiological source of TDO. TDO might support cancer metastasis and progression of disease by affecting anti-cancer immunity or by promoting growth via AHR activation. TDO/IDO inhibitor should have a potential role in improving the efficacy of immunotherapy against MUM and could possibly prolong the survival of MUM patients.

## 4. Materials and Methods

### 4.1. Tissue Specimens

Formalin-fixed paraffin-embedded sections were collected from 16 stage IV UM patients. This study was reviewed and approved by the institutional review board at Thomas Jefferson University (IRB #02.9014R and #11E.548).

### 4.2. Cell Lines and Cell Culture Studies

MUM cell lines (UM001, UM002B, UM004, and UMp005) were established from MUM tissue specimens obtained from UM patients and characterized in our laboratory. Characteristics of these cell lines were previously reported [30,31]. UMp005 was newly established from a PDX mouse model and it was confirmed to have BAP1 mutation at c.828dupT and GNA11 Q209L mutation by Sanger sequencing (Appendix A). UMp005 cells expressed human melanoma markers, including HMB45 and human high molecular weight-melanoma-associated antigen (*HMW-MAA*) detected by flow cytometry (Appendix A). All cell lines were authenticated by short tandem repeat (STR) (ATCC, Manassas, VA, USA). UM001 cells were cultured in RPMI1640 supplemented with 10% fetal bovine serum (FBS), 4 mM L-glutamine, 1% nonessential amino acids, 100 IU penicillin, and 100 µg/mL streptomycin. UM002B and UMp005 cells were cultured with EMEM supplemented with 15% FBS, 100 IU penicillin, and 100 µg/mL streptomycin. UM004 cells were cultured with EMEM supplemented with 10% FBS, 100 IU penicillin, and 100 µg/mL streptomycin. They were confirmed mycoplasma negative determined by MycoProbe^®^ (R&D systems, Minneapolis, MN, USA).

### 4.3. Data Collection from Publicly-Available Datasets

TCGA database contains DNA and RNA-sequence expression data for over 10,000 patients representing 33 distinct cancer types [26,32]. Included in our analyses were 80 primary UM samples. Normalized gene expression data files were downloaded from the genomic data commons (GDC) data portal for 10,258 samples across the 33 cancers. The normalized RNA-sequence as fragments per kilobase of transcript per million mapped reads (FPKM) were used. In addition to these data, clinical information on these 80 UM patients was also retrieved from GDC. This data included all relevant pathological and clinical information and overall prognostic outcomes. Finally, mutation information for the commonly-mutated UM genes (BAP1, GNAQ, GNA11, EIFA1X, and SF3B1) was retrieved. As TDO2 mRNA expression may correlate with somatic copy number alterations (SCNAs) clusters [7], the difference of TDO2 expression among SCNAs clusters was compared. To determine an association between TDO2 expression and BAP1 mutations, we first stratified patients as being BAP1 wild-type versus having a BAP1 alteration. The remaining parameters were stratified based upon being below or above the mean value of the gene. In this manner, the samples were split into two groups and expression of their TDO2 computed in each group. From these two groups, a two-tailed t-test was performed on the normalized TDO2 expression values. Kaplan-Meier analyses were performed to determine OS associations. A log-rank test was used to compute the significance of survival difference.

### 4.4. Immunohistochemistry (IHC)

Formalin-fixed paraffin-embedded (FFPE) tissue sections (4–5 micrometer) were deparaffinized in a series of xylenes and a series of graded ethanol. Antigen retrieval was performed by a heat-mediated method in 10 mM sodium citrate buffer at pH 6.0 for TDO-targeted antigen, in 10 mM citric acid buffer at pH 6.0 for IDO, and in Tris-EDTA buffer at pH 9.0 for HMB45 in a steamer for 20 min. Slides were washed in tris-buffered saline with 0.025% tween 20. Tissue sections were blocked for endogenous peroxidase and alkaline phosphatase by Bioxall (Vector Laboratories, Burlingame, CA, USA) followed by 10% normal horse serum prior to primary antibody incubation for 2 h. Tissue sections were incubated with anti-human TDO (#LS-B5791; dilution 1:200, LifeSpan BioSciences, Seattle, WA, USA), IDO (#D5J4E^TM^; dilution 1:400, Cell Signaling), HMB45 (#M0634; dilution 1:100), or isotype control (both from Aligent, Santa Clara, CA, USA) antibodies overnight at 4 °C. To check the non-specific binding of anti-TDO antibody, an anti-TDO antibody was first incubated with blocking peptides (LifeSpan BioSciences) that interfere with the binding of the anti-TDO antibody to TDO at the final concentration of 1 μg/mL for 30 min at room temperature. The tissue sections were then incubated with peptide-blocked TDO antibody overnight at 4 °C. The primary antibody was washed out and the tissue sections were incubated with ImmPRESS UNIVERSAL reagent (Vector Laboratories) for 30 min, followed by washing with 0.1% tween 20 in tris-buffered saline. The color was developed by ImmPACT VECTOR RED AP (Vector Laboratories). Slides were counterstained with hematoxylin (Vector Laboratories). The intensity of TDO staining of hepatic metastasis was captured using Nikon^TM^ Eclipse 50i microscope with NIS-Elements D3.1 software (Nikon, Melville, NY, USA).

### 4.5. RNAscope^®^ Assay

Tissue sections were also evaluated by TDO2 RNA in situ hybridization with a commercially available kit (Advanced Cell Diagnostics, Inc., Newark, CA, USA) according to the manufacturer’s instructions. Sample quality was validated using probes representing positive (PPIB) and negative (DapB) controls. Briefly, FFPE tissue blocks were sectioned at 5 μm onto Frost Plus slides. Slides were baked for 1 h at 60 °C prior to use. After deparaffinization and dehydration, the tissues were air-dried and treated with peroxidase blocker before boiling for target retrieval in a pretreatment solution for 15 min. Protease was then applied for 30 min at 40 °C. The target probe (TDO2 mRNA) and control probes were hybridized for 2 h at 40 °C, followed by a series of signal amplification and washing steps. Hybridization signals were detected by chromogenic reactions using a red chromogen. Specific RNA staining was identified as red punctate dots. Following the RNAscope staining procedures, samples were counterstained for 2 min with Hematoxylin. Images were captured with a Nikon^TM^ Eclipse 50i microscope with NIS-Elements D3.1 software.

### 4.6. RT-PCR

RNA from the UM cell lines was isolated according to the manufacturer protocol (Qiagen, Valencia, CA, USA). Isolated RNA (2.5 µg) was used for the synthesis of complementary DNA using the Superscript III First-Strand Synthesis System for RT-PCR and Oligo (dT)_20_ primer according to the manufacturer’s protocol (Invitrogen, Carlsbad, CA, USA). Fragments of cDNA were amplified using Platinum Taq DNA Polymerase (Invitrogen). The primers used for TDO2 and IDO1 detection were shown in Appendix A. For TDO2 RNA detection, we used the following conditions: 5 min at 94 °C, 31 cycles (94 °C for 1 min, 56 °C for 1 min, and 72 °C for 1 min), and 10 min at 72 °C amplified to a 601 bp fragment. The IDO1 primers were run in the following conditions: 5 min at 94 °C, 31 cycles (94 °C for 1 min, 61 °C for 1 min, and 72 °C for 2 min), and 10 min at 72 °C amplified to a 1278 bp fragment. The GAPDH primers (Appendix A) were amplified to a 452 bp fragment in the following conditions: 5 min at 94 °C, 30 cycles (94 °C for 30 s, 62 °C for 30 s, and 72 °C for 1 min), and 10 min at 72 °C. Then each reaction mixture and ladder marker were loaded on a 1.7% agarose gel and analyzed with gel electrophoresis. RT-PCR products were visualized using GelRed Nucleic Acid Stai RGB-4103 (Phenix, Candler, NC, USA) under ultraviolet light.

### 4.7. Quantitative PCR

UM cells were seeded at the concentration of 3 × 10^6^ in 100-mm plate and various concentrations of recombinant human (rh) TNF-α, rhIL-6, rhIL-1β, and rhIFN-γ (PeproTech, Inc, Rocky Hill, NJ, USA) or no cytokine was added and incubated for 24 h. Total RNA was collected and complementary DNA was synthesized using SuperScript VIVO (Invitrogen). To determine gene expression level of TDO2 or IDO1, quantitative PCR was performed using PowerUp SYBR Green Master Mix (Applied Biosystems, Foster City, CA, USA), and data analysis was performed using Quant Studio 12K Flex Real-Time PCR instrument (Applied Biosystems). The TDO2 expression was normalized to RNA loading for each sample using GAPDH mRNA as an internal standard. All experiments were performed in triplicate. All PCR conditions were determined by a melting curve analysis.

### 4.8. Western Blotting

Cultured UM cell lines were homogenized in a lysis buffer (Thermo Scientific, Waltham, MA, USA) with a protease inhibitor cocktail. After centrifugation at 12,000 rpm for 15 min at 4 °C, supernatants were collected. Total protein was quantified using the Bradford protein assay (Bio-Rad, Hercules, CA, USA). Protein (20 µg) was separated by a 4–20% SDS-PAGE (Invitrogen) and transferred to PVDF membrane. Membranes were blocked with a blocking buffer (Invitrogen) at room temperature for 1 hr. The membrane was incubated with primary antibodies at 4 °C overnight: mouse anti-human TDO antibody purchased from Abnova (#H00006999-B01P; dilution 1:500, San Jose, CA, USA), anti-human IDO antibody (#D5J4E^TM^; dilution 1:1000) and anti-human β-actin antibody (#D6A8; dilution 1:10,000) purchased from Cell Signaling Technology (Beverly, MA, USA). The membranes were further incubated with secondary antibody, a horseradish peroxidase-conjugated anti-mouse IgG antibody for TDO, or anti-rabbit antibody for IDO and β-actin (both from Cell Signaling) for 1 h at room temperature. Signal was developed using the Westfemto substrate (Thermo Scientific), and image was scanned with C-DiGit blot scanner (LI-CDR, Lincoln, NE, USA).

### 4.9. Detection of Kynurenine (Kyn) Levels in Culture Supernatants

UM004 cells were seeded at 2.5 × 10^5^ cells per well with 500 μL culture medium without FBS in a 48-well plate. The culture medium was harvested after 24 h incubation, centrifuged, and frozen until further analysis. Original culture medium and standards with known concentration of Kyn were used for standardization of assay. Kyn reference standards were obtained from Sigma-Aldrich Corp (St. Louis, MO, USA). HPLC-MS-grade acetonitrile (ACN), HPLC-MS grade water (with and without 0.1% formic acid (FA)), and FA were purchased from Fisher Scientific (Fair Lawn, NJ, USA).

Reverse-phase, isocratic elution chromatography was performed using HPLC-MS (Thermo Fisher, Waltham, MA, USA), for detection and quantitation of Kyn. The mass spectrometer was calibrated in positive ion mode with a mass error of 0.2 to 0.3 parts per million (ppm). The scanning mass range was set as 190 to 211 m/z. This mass spectrometer method was used for the detection of Kyn eluted from Dionex Ultimate 3000 HPLC system. Separation of Kyn was carried out on C_18_ reverse phase (3.5 µm particle size, 4.6 × 75 mm) XBridge column by Waters (Milford, MA, USA). Standards and samples were eluted using 20% ACN with 0.1% FA in distilled water. The injection volume was set to 2 µL with a solvent flow rate of 0.350 mL/min. The column temperature was set to 30 °C. Thermo XCaliber (v. 3.0.63) and Exactive (v. 1.1SP6) software were used for method development and data acquisition in raw format (Appendix A).

For quantitative estimations, a standard curve was developed in the range of 2 ng/mL to 2000 ng/mL of Kyn in 20% ACN with 0.1% FA in distilled water. The standard curve was replicated on three different days to determine %CV and recovery (Appendix A). Standards and samples were thawed and kept at 20 °C. The collected medium was added into ACN with 0.1% FA at the ratio of 1:1. The samples were vortexed for 40–50 s, centrifuged at 15,700× *g* for 15 min, and then 150 µL supernatant was taken and analyzed.

### 4.10. Statistical Analysis

The data were expressed as the mean ± standard deviation. Differences between the groups were analyzed using the Student’s *t*-test. A *p*-value < 0.05 was considered significant. Kaplan–Meier analyses were performed for survival analysis. A log-rank test was used to compute the significance of survival difference.

## 5. Conclusions

Our results suggest that liver metastasis from uveal melanoma express TDO and produce Kyn in the tumor microenvironment. The expression of TDO in the liver metastasis may facilitate inhibition of anti-tumor immunity and TDO may be an important target to improve the efficacy of immunotherapies against MUM in the liver.

## Figures and Tables

**Figure 1 cancers-12-00405-f001:**
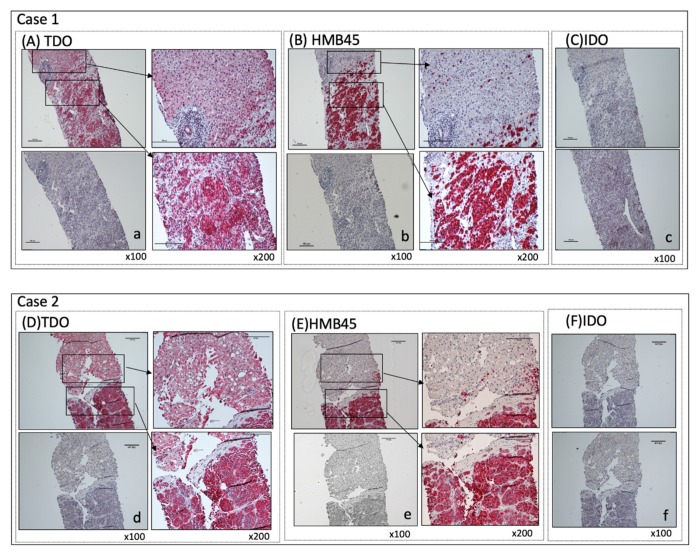
Immunohistochemical (IHC) staining of liver metastasis specimens. Liver metastasis specimens were stained for TDO, HMB45, and IDO. Positive cells show a distinct red stain. IHC staining of two representative sets of archived liver metastasis specimens were shown. (**A**,**D**) TDO staining: For negative control, TDO antibody was blocked with specific blocking peptide (a,d). Magnification: ×100 (left panels); ×200 (right panels) Scale bar = 100 μm. (**B**,**E**) HMB45 staining: Melanoma metastasis is positive (right bottom) and surrounding liver tissue is negative (right top). Isotype-matched IgG was used as negative control (b,e). Magnification: ×100 (left panels); ×200 (right panels) Scale bar = 100 μm (**C**,**F**) IDO staining: IDO is negative for both metastatic melanoma and surrounding liver tissue (top). Isotype-matched IgG is used as negative control (c,f). Magnification: ×100. Scale bar = 100 μm.

**Figure 2 cancers-12-00405-f002:**
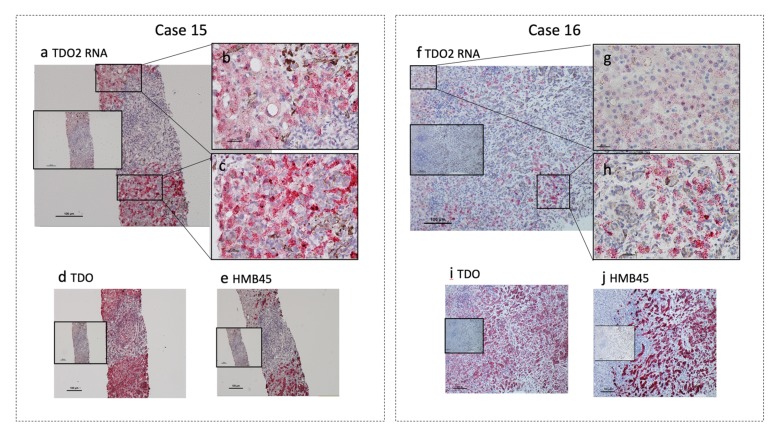
Localization of TDO2 RNA and TDO protein in hepatic metastasis specimens (Case 15 and 16). In situ hybridization of TDO2 RNA (**a**–**c**,**f**–**h**): Positive cells with TDO2 RNA show a distinct red stain. Metastatic uveal melanoma cells show stronger and distinct staining (**c**,**h**) compared to the surrounding liver (**b**,**g**). Staining with unrelated control probe is shown as negative control in the mid-left inserts. TDO protein staining (**d**,**i**): Areas positive for in situ staining of TDO2 RNA are also strongly positive for TDO protein. Staining with isotype-matched control antibody is shown in the mid-left inserts. HMB45 staining (**e**,**j**): The areas strongly positive for TDO2 RNA (**c**,**h**) and TDO protein are also positive for HMB45. Neighboring liver tissue with much weaker staining for TDO2 RNA (**b**,**g**) and TDO protein is negative for HMB45. Staining with isotype-matched control antibody is shown in the mid-left inserts. Magnification: ×100 (**a**,**d**,**e**,**f**,**i**,**j**) Scale bar = 100 μm; ×400 (**b**,**c**,**g**,**h**) Scale bar = 20 μm.

**Figure 3 cancers-12-00405-f003:**
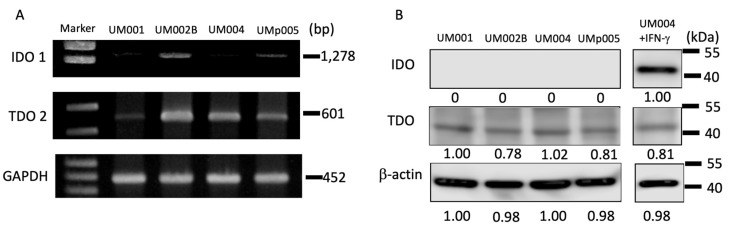
Expressions of TDO and IDO in MUM cell lines. (**A**) IDO1 and TDO2 mRNA expressions were evaluated by RT-PCR. GAPDH was used as a loading control. (**B**) TDO and IDO protein expression in MUM cell lines. β–actin was used as a loading control.

**Figure 4 cancers-12-00405-f004:**
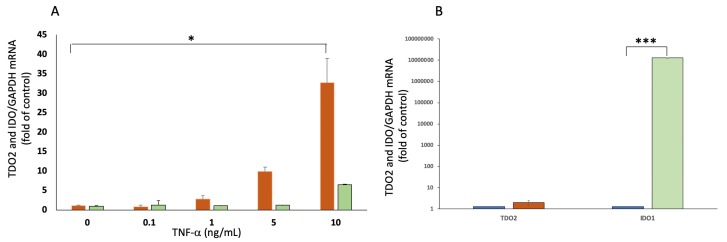
Expressions of TDO2 RNA and IDO1 RNA in response to TNF-α and IFN-γ. (**A**) qPCR analysis of TDO2 RNA expression in UM004 cells treated with various concentrations of TNF-α for 24 h. * *p* < 0.05, compared to baseline. 
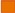
: TDO2 mRNA, 
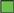
: IDO1 mRNA. (**B**) The expression of TDO2 RNA and IDO1 RNA in UM004 cells cultured with 1 ng/mL of IFN-γ for 24 h. *** *p* < 0.001, compared to control (no IFN-γ). 
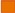
: TDO2 mRNA, 
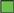
: IDO1 mRNA.

**Figure 5 cancers-12-00405-f005:**
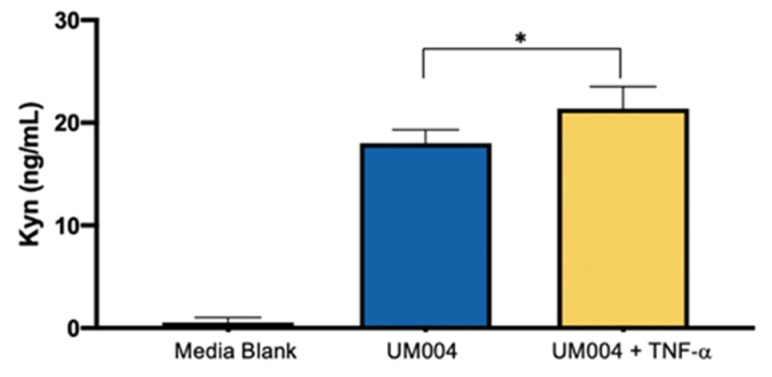
Detection of Kyn in culture supernatants from UM004 cells. MUM cell line (UM004) was cultured with or without 10 ng/mL of TNF-α in serum-free medium for 24 h. The concentration of Kyn is measured using LC-MS. Kyn production was increased by TNF-α (* *p* < 0.05, compared to no addition of TNF-α).

**Figure 6 cancers-12-00405-f006:**
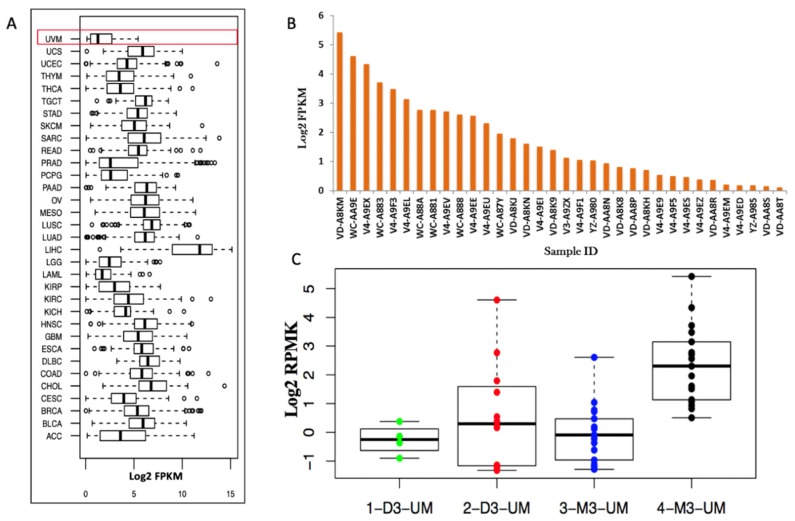
TDO2 expression in primary uveal melanoma. (**A**) Identification of TDO2 RNA expression across other cancer types in the TCGA database. The first box shows UM data. The *y*-axis is a type of cancer (Abbreviation in Appendix A). The *x*-axis is a log2 FPKM values. (**B**) 34 primary uveal melanoma tissue samples with positive TDO2 RNA. The *x*-axis shows individual IDs from TGCA data. The *y*-axis is a log2 FPKM values. (**C**) TDO2 mRNA expression, grouped by somatic copy number alterations (SCNAs) clusters. Dots show all data values. The *y*-axis is a log2 FPKM values for TDO2 RNA. Box plots show median values, and the 25th to 75th percentile range in the data, i.e., the interquartile range (IQR).

**Figure 7 cancers-12-00405-f007:**
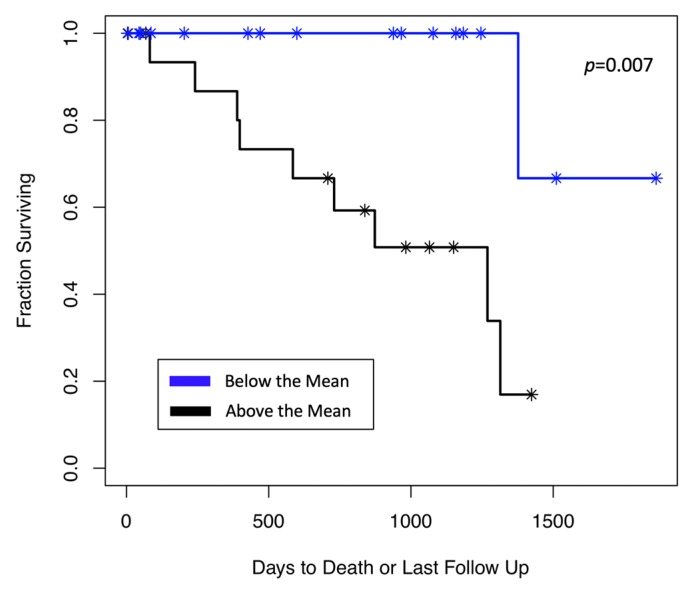
TDO2 RNA expression and survival of patients in primary UM. Patients with a high TDO2 RNA expression had shorter survival compared to those with low TDO2 RNA expression (*p* = 0.007).

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
