# Peer review of "Expression of Tryptophan 2,3-Dioxygenase in Metastatic Uveal Melanoma"

_cancers, 2020, doi:10.3390/cancers12020405_

Round 1

Reviewer 1 Report

This manuscript examines the expression of TDO and IDO in uveal melanoma (UM) cells based on the hypothesis that extracellular kynurenine (kyn) produced by UM tumors inhibits immune cell response during immune checkpoint therapy. The need for successful therapeutic options to treat metastatic UM is paramount; so, identifying why UM tumors do not respond to immunotherapy could uncover new treatment strategies. However, there are several issues that need to be addressed.

The authors state that “PD-L1 can be expressed in metastatic UM cell lines in response to interferon-gamma (IFNγ) and this phenomenon indicates lack of T cell attack in MUM tumor microenvironment” (Line 53) as part of their justification for evaluating TDO and IDO as modifiers of the microenvironment that could affect T cell activation. However, they show in Figures 3 and 4 that IDO is strongly upregulated in UM cells in response to IFNγ. If IFNγ induction of PD-L1 is important, then the effect of IFNγ on IDO function must be characterized. The authors need to perform the same LCMS media kyn measurements presented in Figure 5 but with IFNγ. This is vital to answering whether release of IFNγ by activated T-cells in the tumor would then induce IDO expression to quench the activation. This could potentially be a more important mechanism of immune evasion than simple expression of TDO. TDO2 levels are significantly higher in cutaneous melanoma (Figure 6) than uveal melanoma; however, metastatic cutaneous melanomas show some of the strongest responses to immune checkpoint therapy. The authors should discuss why these higher levels of TDO expression are not protective in other melanomas. “Trp” is the accepted IUPAC abbreviation for tryptophan and should replace “Tryp” everywhere in the manuscript. In Figures 1 and 2, the panel labels (a, b, c, etc.) should all be replaced with TDO, HMB45, IDO and “liver”, “tumor” for clarity. The reader should be able to follow the figures without constantly referring to their legends. As they stand, the figures do not fully convey what the authors intend in the text. In Figure 1, the isotype control insets in the top set of panels are impossible to see even zoomed in and should be the same size as their accompanying images. All of the panels in this set are too small to see any actual histology making it difficult to appraise cytology, localization, or intensity. These images should be presented at high resolution and placed in the supplemental figures with two representative sets (high expressing and low expressing) presented in Figure 1. Table 1 is missing a title and legend.

Author Response

Response to Reviewer 1

(1) The authors state that “PD-L1 can be expressed in metastatic UM cell lines in response to interferon-gamma (IFNγ) and this phenomenon indicates lack of T cell attack in MUM tumor microenvironment” (Line 53) as part of their justification for evaluating TDO and IDO as modifiers of the microenvironment that could affect T cell activation. However, they show in Figures 3 and 4 that IDO is strongly upregulated in UM cells in response to IFNγ. If IFNγ induction of PD-L1 is important, then the effect of IFNγ on IDO function must be characterized. The authors need to perform the same LCMS media kyn measurements presented in Figure 5 but with IFNγ. This is vital to answering whether release of IFNγ by activated T-cells in the tumor would then induce IDO expression to quench the activation. This could potentially be a more important mechanism of immune evasion than simple expression of TDO.

Thank you for this comment.  The upregulation of IDO and PD-L1 in response to IFNγ are well known phenomena and this is the key rationale in using an IDO inhibitor in combination with an immune checkpoint inhibitor.  Since IFNγ does not induce upregulation of TDO in MUM cells and IDO is not expressed in metastatic uveal melanoma tissues, it is not within our scope of current research to investigate the role of IFNγ in MUM cell lines including kyn measurements.  We have added the sentences to clarify this point [lines 167-168].

 (2) TDO2 levels are significantly higher in cutaneous melanoma (Figure 6) than uveal melanoma; however, metastatic cutaneous melanomas show some of the strongest responses to immune checkpoint therapy. The authors should discuss why these higher levels of TDO expression are not protective in other melanomas.

Thank you for this valuable comment.  We investigated skin melanoma specimens tested in TCGA and confirmed that the majority of specimens are derived from metastasis. TCGA SKCM datasets consist of 103 thick primary samples (21.9%) and 368 metastatic samples (78.1%).  On the other hand, in the TCGA UVM datasets, all data are obtained from primary samples.  Therefore, we do not believe that the comparison of skin and uveal melanoma TCGA data is meaningful. 

Furthermore, PD-L1 expression and the response to immune checkpoint inhibitors tend to be low even in metastatic skin melanoma if metastasis developed in the liver [Javed  et. al.,  Immunotherapy (2017) 9: 1323–1330;  Lee et. al. JCO (2017) 35, 15_suppl: 3072-3072.].  This indicates that tumor microenvironment in the liver (including TDO) might also play a role in suppressing the response to immune checkpoint inhibitors, although this is solely speculative.  We agree that further investigation of TDO expression in metastatic skin melanoma is needed before making any definitive conclusion.

(3) “Trp” is the accepted IUPAC abbreviation for tryptophan and should replace “Tryp” everywhere in the manuscript.

Thank you for this correction.  We have revised the word “Tryp” to “Trp” in our manuscript.

(4) In Figures 1 and 2, the panel labels (a, b, c, etc.) should all be replaced with TDO, HMB45, IDO and “liver”, “tumor” for clarity. The reader should be able to follow the figures without constantly referring to their legends. As they stand, the figures do not fully convey what the authors intend in the text. In Figure 1, the isotype control insets in the top set of panels are impossible to see even zoomed in and should be the same size as their accompanying images. All of the panels in this set are too small to see any actual histology making it difficult to appraise cytology, localization, or intensity. These images should be presented at high resolution and placed in the supplemental figures with two representative sets (high expressing and low expressing) presented in Figure 1.

Thank you for the suggestions.  We have changed the labels according to this suggestion. We have placed two representative images in Figure 1 and the remaining images are placed in the supplement figure [Figure S1].  We did not quantify the degree of TDO protein levels in individual specimens; therefore, the representative tissue specimens containing tumor and surrounding liver tissue are shown for internal comparison. 

(5) Table 1 is missing a title and legend.

We have included the results shown in Table 1 in the text and deleted Table 1.

Reviewer 2 Report

Terai M and colleagues proposed a very interesting article aimed at elucidating the role of tryptophan 2,3-dioxygenase (TDO) in suppressing the anticancer property of immunotherapy inhibiting the activation of T-cell in metastatic uveal melanoma. The authors analyzed both in silico, in vitro and in vivo data highlighting the over-expression of TDO (but not of IDO) in both FFPE MUM tissues and MUM cell lines. The authors analyzed only a limited number of FFPE samples (n. 10) obtained from liver metastasis. It would have been appropriate to extend the analysis to more samples including also FFPE related to primary UM. Overall, the manuscript is well written and the data obtained are convincing. Below are reported some minor/major revisions that will improve the quality of the manuscript:
1) In paragraph 2.3, and in the entire manuscript, please substitute “TDO2 message” with “TDO2 mRNA”;
2) In the subheading “4.1. Tissue specimens” please specify the number of FFPE sections analyzed;
3) In the subheading “4.3. Data Collection from publicly-available datasets” the authors state “Primary uveal melanoma patients for which TDO2 was not expressed were removed from further analysis, leaving a total of 34 patients to be analyzed for clinical correlation”. I suggest to the authors to perform further statistical analyses dividing the UM patients into two different groups: Group 1 patients overexpressing TDO and group 2 patients with normal TDO expression. Have the authors performed correlation and survival analyses taking into account these two groups? Please clarify;
4) In the paragraph “4.8. Western Blotting” of the Materials and Methods section please specify the catalog number and the concentration of the antibodies used;
5) How the authors explain the constitutive absence of IDO protein expression in both FFPE specimens and MUM cell lines? Please clarify this point in the discussion section;
6) It is not clear why the authors stimulate MUM cells with TNF-α and IFN-γ. Please describe better the reason of such experiments in the Materials and Methods section;
7) As previously mentioned, it would have been appropriate to extend the analysis to more samples including also FFPE related to primary UM. Please consider also primary UM in your future experiments (this is only a suggestion).

Author Response

(1) In paragraph 2.3, and in the entire manuscript, please substitute “TDO2 message” with “TDO2 mRNA”;

We have changed the word “TDO2 message” to “TDO2 mRNA”.

(2) In the subheading “4.1. Tissue specimens” please specify the number of FFPE sections analyzed;

We have stained a total of 16 hepatic metastasis samples. We have added the number of sections (patient specimens) in the text line [83] and [318].

(3) In the subheading “4.3. Data Collection from publicly-available datasets” the authors state “Primary uveal melanoma patients for which TDO2 was not expressed were removed from further analysis, leaving a total of 34 patients to be analyzed for clinical correlation”. I suggest to the authors to perform further statistical analyses dividing the UM patients into two different groups: Group 1 patients overexpressing TDO and group 2 patients with normal TDO expression. Have the authors performed correlation and survival analyses taking into account these two groups? Please clarify;

Thank you for this valuable suggestion.  We did perform survival analyses comparing those subjects with positive TDO2 expression (n=34) vs. those without TDO2 expression (n=46).  Interestingly, in this analysis, there was no difference in survival among these two groups.  It could be speculated that TDO does not have any meaningful role in tumors that do not express TDO.  There are many different immune suppressive mechanisms in cancer cells and a different immuno-modulatory mechanism would exist in primary UM without TDO expression. Alternatively, it would also be possible that UM starts expressing TDO in response to a factor in tumor microenvironment. We identified one possible factor as TNF-a.  Furthermore, TDO2 mRNA is dominantly expressed in Cluster 4 primary UM in which significant infiltration of immune cells are present.  We agreed that further investigation is needed for better understanding of the mechanism of TDO2 expression in MUM and added the above speculation in the revised text [lines 246-259].

We also compared survival within the patients who had constitutive TDO2 expression in their primary uveal melanoma (n=34).  We grouped them based upon median expression of TDO2 (those above the median vs. those below the median).  In this scenario, there was a clear association between TDO2 expression and survival of UM patients.  There was a p-value of 0.007. The degree of TDO expression is correlated to poor prognostic markers such as BAP-1 mutation and monosomy 3.  These findings might also indicate an unique biological role of TDO in primary uveal melanoma.  We included these findings in [lines 196-203] and added Figure [7] for clarification.   

(4) In the paragraph “4.8. Western Blotting” of the Materials and Methods section please specify the catalog number and the concentration of the antibodies used.

 We have inserted the catalog number and dilution in the section of Western Blotting in the Material and Methods.

(5) How the authors explain the constitutive absence of IDO protein expression in both FFPE specimens and MUM cell lines? Please clarify this point in the discussion section.

Our results indicate that there is no constitutive expression of IDO in metastatic uveal melanoma, which is not unusual in many other types of cancer [https://www.proteinatlas.org/ENSG00000131203-IDO1/pathology]. 

It is also known that IDO is inducible in cancer cells in response to IFNγ.  In fact, MUM cell lines are able to upregulate IDO when co-cultured with IFNγ.  Based on these data, we speculated that adequate amounts of IFNγ are not present in the tumor microenvironment in metastatic UM.  It is not within the scope of our current research to compare IDO and TDO expression in MUM; however, positive expression of TDO and negative expression of IDO in all MUM tissues and cell lines might indicate a dominant role of TDO in MUM.

(6) It is not clear why the authors stimulate MUM cells with TNF-α and IFN-γ. Please describe better the reason of such experiments in the Materials and Methods section;

Thank you for this comment.  In fact, we tested several pro-inflammatory cytokines, including IL-6 and IL-1beta.  However, none of these cytokines upregulated expression of TDO or IDO, except TNF-alpha.  Since IFNγ is known to upregulate IDO, we also included the data on IFNγ stimulation as comparison to TNF-alpha. We have included the types of cytokines that we tested for screening in the text [line 147-148].  

(7) As previously mentioned, it would have been appropriate to extend the analysis to more samples including also FFPE related to primary UM. Please consider also primary UM in your future experiments.

All patients whose metastases were analyzed in this study received radioactive plaque treatments; therefore, their primary uveal melanoma tissues are not available.  It would be very interesting to test primary uveal melanoma for TDO expression; however, only 10-15% of patients currently undergo enucleation of their eye and these patients might not represent the total UM patient population.  The majority of UM patients are currently treated with radioactive plaque and their biopsy specimens are limited.  Although the suggestion by this reviewer is very important, we would not be able to conduct a conclusive study on primary UM. 

Round 2

Reviewer 1 Report

While the question of lack of IDO expression in liver mets despite its response to IFNg is still not adequately addressed, the manuscript is much improved overall. There is now sufficient discussion of the dichotomies of TDO and IDO expression in primary versus metastatic samples to cover initial concerns. The authors have addressed all of the other issues very well and the manuscript is much improved.